# Long-Term Mental Health Support after Natural Hazard Events: A Report from an Online Survey among Experts in Japan

**DOI:** 10.3390/ijerph19053022

**Published:** 2022-03-04

**Authors:** Ryoma Kayano, Mingming Lin, Yasuko Shinozaki, Shuhei Nomura, Yoshiharu Kim

**Affiliations:** 1Centre for Health Development, World Health Organization, 1-5-1 Wakinohama-Kaigandori, Chuo-ku, Kobe 651-0073, Japan; 2Laboratory for Imagination and Executive Functions, RIKEN Center for Brain Science, Wako-shi 351-0198, Japan; mingming.lin@riken.jp; 3Mental Health and Disability Health Division, Department of Health and Welfare for Persons with Disabilities, Social Welfare and War Victims Relief Bureau, Ministry of Health, Labour and Welfare, 1-2-2 Kasumigaseki, Chiyoda-ku, Tokyo 100-8916, Japan; shinozaki-yasuko.6h3@mhlw.go.jp; 4Department of Health Policy and Management, School of Medicine, Keio University, 35 Shinanomachi, Shinjuku-ku, Tokyo 160-8582, Japan; s-nomura@keio.jp; 5Department of Global Health Policy, Graduate School of Medicine, The University of Tokyo, 7-3-1 Hongo, Bunkyo-ku, Tokyo 113-0033, Japan; 6Tokyo Foundation for Policy Research, 3-2-1 Roppongi, Minato-ku, Tokyo 106-6234, Japan; 7National Institute of Mental Health, National Center of Neurology and Psychiatry, 4-1-1 Ogawahigashi, Kodaira 187-8553, Japan; kim@ncnp.go.jp

**Keywords:** health EDRM, long-term, disaster mental health, natural hazards, Japan, survey

## Abstract

This paper aims to provide preliminary evidence on the degree of consensus on the approach to long-term mental health and psychosocial support after a natural hazard event. We conducted an online survey among mental health experts in Japan. The questionnaire was divided into five categories: (A) terminology setting definition of “long-term”, (B) priority in activity for long-term mental health support, (C) system and preparedness for better support, (D) transition from acute support to long-term support, and (E) actions to improve preparedness for future disasters. Invitations to participate in the survey were sent by e-mail in November 2017 to mental health experts in Japan, who had participated in workshops related to disaster mental health or trauma care organized by the National Institute of Mental Health over the last 15 years. Out of 1385 experts who received the invitation, a total of 305 participants responded to the survey. Participants were for the most part in agreement regarding focuses and required preparedness and actions for long-term support. There was still low consensus especially on defining the timeframe “long-term”. The acute phase and long-term phase were identified as being different in dimension rather than category. Although caution is necessary around the representativeness of these findings, they will provide important scientific evidence for the development of future plans for a qualitative improvement in long-term mental health support.

## 1. Introduction

Emergencies and disasters typically place a significant and persistent mental health burden on those directly and indirectly affected, as well as those who respond to the demand for their services. The need for mental health and psychosocial support (MHPSS) is also highlighted in the Sendai Framework for Disaster Risk Reduction 2015–2030 [1]. A number of studies as well as international guidelines and manuals for psychosocial support in disasters have been published (e.g., The Sphere Project [2]; Inter-Agency Standing Committee [3]). However, in most previous literature the main focus has been put on the acute phase; long-term MHPSS are only outlined and lack precise and concrete principles and strategies. In addition, according to a recent systematic review that assessed the long-term health consequences of disasters, the health outcome that was assessed the most was mental disorders [4], suggesting providing MHPSS [5,6]. However, while most literature shares the basic principle that the long-term phase requires community-based MHPSS interventions, a precise description of the interventions is scarce in contrast to the rich and abundant suggestions for activities in the acute phase [7,8,9].

As one of the most disaster-prone countries in the world [10], but also one of the most prepared due to its significant exposure to disaster risks, Japan may have a relatively large body of knowledge and lessons on long-term disaster response among many disaster-prone countries [4]. In Japan, the importance of long-term MHPSS interventions was first recognized in Japanese literature after the volcanic eruption of Mt. Unzen-Fugen in 1991, where victims had to be relocated to distant areas far from their homeland [11,12]. In 1995, there was the Great Hanshin and Awaji Earthquake in Hyogo Prefecture that caused nearly 6000 deaths [13]. For the first time in Japanese history, the affected local government decided to establish the Hyogo Institute for Traumatic Stress [14], a local mental healthcare center in the Hyogo Prefecture dedicated to the care of the victims’ mental health. The institute continues to provide MHPSS in the community, as well as education for professional staff and the treatment of severely traumatized victims. This model of long-term MHPSS based on a local mental healthcare center was repeated later in large natural hazards such as the 2004 Chuetsu Earthquake, the 2011 Great East Japan Earthquake and Tsunami disasters, and the 2016 Kumamoto Earthquake [15,16]. In other disasters where such dedicated mental healthcare centers were not established, most local prefectures and mental health and welfare centers in the disaster sites continued to follow up on the victims [17].

MHPSS interventions in a disaster context should have the clear vision that the long-term community-based mental healthcare is critically important to promote the psychosocial recovery and quality of life of the victims [18,19]. However, there is no global consensus regarding long-term MHPSS after a disaster, although there are standard MHPSS manuals or systems for the acute phrase (e.g., Disaster Psychiatric Assistance Team (DPAT) [20]; Psychological First Aid, World Health Organization [21]). For the long-term phase, although there are event-based practical reports from local communities or mental healthcare facilities from the areas affected by natural hazard events, as well as several guidelines or manuals by various academic societies and local or national facilities (e.g., National Information Center of Stress and Disaster Mental Health, National Center of Neurology and Psychiatry, Japan), there is still an absence of an internationally accepted standard. There is also a lack of consensus over the definition of the timeframe for acute- and long-term recovery after disasters. It is crucial to elucidate whether Japan’s accumulated experience in MHPSS interventions after disasters ever created professional consensus regarding long-term MHPSS. It is also important to discuss how to integrate the consensus or overcome dissidence to establish a common understanding or guiding principles.

This paper aims to provide preliminary evidence of the degree of consensus on the approach to long-term MHPSS interventions after a natural hazard event. We conducted an online questionnaire survey among mental health professionals in Japan. To our knowledge, this is the first attempt to investigate experts’ consensus view on long-term MHPSS after natural hazard events. Based on the above-mentioned background of Japan, opinions provided by mental health experts in Japan could provide important scientific evidence for future plans of support for a comprehensive mental health policy related to natural hazard events.

## 2. Materials and Methods

### 2.1. Participants and Procedures

An online questionnaire survey was conducted to investigate the cohesiveness and dissidence in opinions on long-term MHPSS after a disaster. The questionnaire was developed through SurveyMonkey (a widely-used free online survey tool [22]), and the response data were collected anonymously. The language used in the survey was Japanese. Invitations to participate in the survey were sent by e-mail to 1814 mental health experts who had ever participated in workshops related to disaster mental health or trauma care organized by the Japanese National Institute of Mental Health (NIMH) over the last 15 years, and who had registered their e-mail address. Overall, 429 emails returned an error message due to invalid email addresses, leaving 1385 individuals who received the e-mail. A total of 305 participants agreed to participate and responded to the survey (response rate 22.0%) between 2 and 21 November 2017. The response data were anonymous. This survey was conducted as part of NIMH’s efforts to obtain feedback from workshop participants.

### 2.2. Questionnaire

A 28-item questionnaire was developed to assess how much people agree to each statement about long-term MHPSS after a hazard event, using a 6-point Likert scale (0 = disagree; 5 = agree). The questionnaire was divided into 5 categories: (A) terminology setting definition of “long-term” (6 items), (B) priority in activity for long-term mental health support (9 items), (C) system and preparedness for better support (5 items), (D) transition from acute support to long-term support (2 items), and (E) actions to improve preparedness for future disasters (6 items). Content for the questionnaire was developed based on the review of existing literature and consultations with experts. Furthermore, the survey also contained questions about participants’ demographic characteristics and expertise in disaster mental health activity.

## 3. Results

Demographic data are shown in Table 1. Of the 305 participants, 93 (30.2%) were psychologists, 50 (16.4%) were medical doctors, 50 (16.4%) were nurses, and 37 (12.1%) were psychosocial workers. Just over half (153, 50.2%) of the participants worked for medical facilities, 68 (22.3%) worked in administrative offices, and 35 (11.5%) in schools. The largest number of participants (37.7%) were located in Kanto (i.e., the region near the capital of Japan), while others were distributed across the country. Participants were therefore from multiple workplaces and occupations, and from various geographical regions.

The descriptive statistics for the response results (average, standard deviation, maximum/mininmum value, and number of respondents) are shown in Table 2. Regarding the definition of “long-term” (Category A), item no. 1 relates to the statement “Acute stress disorder or traumatic response ends as a natural recovery process (about 1–2 months),” and showed the lowest consensus in this category. The next lowest average was item no. 5 which related to activities based on the Disaster Relief Act. Items no. 1, no. 2, and no. 6, especially, showed a higher standard deviation than the ones in other categories, which may confirm the difficulty and complexity of defining this term. Higher consensus on the definition for “long-term” was observed when related to the recovery of infrastructure and local mental healthcare centers (no. 3 and no. 4).

Results of Category B relate to the priority given to activity for long-term mental health support, and showed that the proposed focuses were highly supported by the participants—except for the activity on diagnostic evaluation and treatment for psychological disorder (no. 7). The highest consensus was observed in no. 10, referring to collaboration between mental healthcare providers and national/local municipalities. Regarding the system and preparedness for better support (Category C), all proposed focuses showed high consensus, with the highest being for the large-scale provision of training programs with the capacity to build on long-term mental health support.

In the two items of Category D which relate to the transition from acute support to long-term support, item no. 21, showed the lowest consensus among all items. Item no. 21 states that the acute phase and long-term phase supports should be organized separately, and that long-term support should be initiated after the termination of acute support. In contrast, the statement that acute- and long-term support should be organized in parallel and in collaboration (item no. 22), showed high consensus. Lastly, all the items in Category E related to actions to improve preparedness for future disasters were highly supported by the participants of the survey. The highest consensus was observed in the training of long-term mental health support for healthcare workers engaged in acute response (item no. 24) and the need to broadly accumulate and inherit past experience for disaster responses (item no. 26). In both items, no one among the respondents gave a score of 0 or 1.

## 4. Discussion

This study provides preliminary evidence for the degree of consensus on a strategy for long-term mental health support after a natural hazard event. Participants were asked to share their ideas on how much they agree to 28 statements on long-term MHPSS after a natural hazard event. While there was a high degree of consensus on many statements, there was also a low degree of consensus on others.

Recent international literature and guidelines emphasize the distinction between mental health activities in the acute- and long-term recovery phase after a natural hazard event, but in practice there is poor consensus regarding the definition of “long-term”, as the psychological effects of the disasters were considered likely to persist for months to even years [4,23]. Indeed, a literature review by Lorenzoni et al. (2020) found that studies assessing the long-term impact of disasters on the public health system as a whole usually failed to provide a rigorous discussion, definition, or rationale for the definition of “long-term” [24]. Thus, we did not request for information on the time period itself but information on the quantitative definition of “long-term”, however, a similar discrepancy was reproduced (Category A). All the items in this category, when compared to the other categories, showed relatively poor consensus and greater standard deviation, which indicates the difficulty and complexity of defining this term even among experts, in spite of similar domestic experiences in disaster response. Item no. 1 showed the lowest consensus, indicating that the end of acute stress disorder as a disease category is not regarded as a good index to differentiate the time period of the post-disaster situation in the community. This may reflect a general tendency to discuss post-disaster MHPSS from a public health viewpoint in the affected communities rather than a narrow medical model to treat specific disorders [25]. Similarly, Seto et al. (2019) argued that there is also a lack of consensus on the definition of the acute phase of a disaster, and the MHPSS needs in the acute phase may vary across affected communities [26]. In addition, the diagnostic period of acute stress disorder of 1 month [27] might seem too short for actual natural hazard event recovery. Item no. 2, “disaster period is over”, might be difficult to interpret, because most frequent disasters in Japan are earthquakes and they are usually followed by minor seismic activity that make it difficult to declare a termination [28]. Contrary to our expectation, the activities based on the Japanese Disaster Relief Act (i.e., a major law that deals with emergency relief in the acute phase after a disaster occurs [29]) were not highly evaluated as a practical initiation or operation for long-term mental health support (item no. 5). This was surprising because most of the responders were public servants and work within the framework of the law. The background may be that this Act regards the appeal for support from other prefectures [30] and ends when the support resources in the affected site are recovered, which does not necessarily synchronize with the needs of the affected victims and community.

Items no. 3, no. 4 and no. 6 regarding the external environment (infrastructure, local mental healthcare center, and temporary houses) marked higher consensus with lower standard deviation, except for no. 6. In particular, the functional recovery of local mental healthcare centers was highly rated by the participants as the character representing “long-term.” These items emphasize the importance of the social network in the definition of long-term support. Item no. 6 marked a lower consensus than no. 3 and no. 4; this may be because the transition to temporary houses is literally a temporary residence and the community has not yet been re-established [31,32,33].

The result of the Category B, “priority in long-term support” showed fairly unanimous consensus regarding public health and social support (items nos. 8–15) while the item for medical support (item no. 7) showed poorer consensus. Here, as discussed above, most participants might consider long-term MHPSS as a public health issue rather than the treatment of specific disorders [25]. This seems to be particularly important, as 50% of the participants worked in medical institutions, and 40% were medical doctors and nurses, and yet they still preferred the psychosocial approach. The service for PTSD and grief mentioned in item no. 15 was also highly rated, presumably because the item used the expression “care system”, and not medical treatment, and that trauma and grief are conceived as psychological states rather than mental disorders [34]. Previous research has also found that social interventions including the strengthening of social connectedness, social networks as well as perceived social support [35], protect against both post-traumatic stress and general long-term mental health problems after disasters [36,37].

Most of items in Categories C, D, and E, “the system and preparedness for better support”, “the transition from acute support to long-term support”, and “the actions to improve preparedness for future disasters”, respectively, were highly rated. Item no. 21, which recommends the separation of acute medical and long-term psychosocial support, showed the lowest level of consensus. This may highlight the consensus for the integrity of both supports. Combined with the results from the previous category, these results show high consensus among the respondents, which may suggest prioritizing seamless public health support (from acute to long-term) through collaboration among multiple experts (mental, health, and social support) rather than dividing those two phases chronologically or by type of support (medical support for acute phase and public health support for long-term phase) [38]. Furthermore, these mental health experts concurred on the need for an evidence-based and expanded capacity of preparation for service providers for long-term psychosocial support for disaster survivors [39].

The fact that most components regarding focuses and required preparedness and actions for long-term support had high consensus in the survey indicates that the experts in post-disaster MHPSS have succeeded in a level of consensus building. This level of consensus is thought to contribute to national initiatives and guidelines, continuous provision of workshops and training programs, efforts by academic societies, and national and local initiatives on major natural hazard management in Japan (e.g., the Hanshin Awaji Earthquake, the Great East Japan Earthquake). However, the survey noted some difficulty in reaching a consensus for defining of the timeframe of “long-term.” The acute phase and the long-term phase are identified to be different in dimension, rather than category, since those experts reached a higher consensus on what is required for long-term support. This is a required framework for effective long-term post-disaster MHPSS with terminology settings on “long-term” and related activities.

The strength of this study is that we were able to use the network of the Japanese National Institute of Mental Health to approach diverse experts in MHPSS in Japan. However, this study has several limitations. Some of the items in the questionnaire lack detailed definitions, for instance “local mental healthcare centers.” Additionally, in this study, we only evaluated the degree of consensus for each of 28 statements with the mean of a 6-point Likert scale response among the respondents, but we did not test whether there was a statistically significant difference in the degree of consensus between each statement, as significance tests, which are influenced by the sample size, were not within the scope of the study. In addition, in general, categorizing information (into six options) means that information within a category is lost, and everyone above or below the cut point is treated equally. The degree of consensus can vary greatly without each category. It should also be noted that this study provides preliminary evidence of the degree of consensus among a limited number of experts in Japan, and that the results are not necessarily representative of Japanese experts, as there is a significant selection and sampling bias involved in the way that the participants in this study were recruited. For example, whether or not people participate in our survey may depend on their interest in the study scope. A previous study has also suggested that online surveys may result in lower risk perceptions of major environmental challenges (including earthquakes) compared to interviews or other survey methods [40]. This bias may have influenced the questions about the urgency of mental health support and its enhancement after natural hazard events. Other selection and sampling biases commonly observed in online surveys also need to be recognized [41]. It is also possible that the results may differ from the consensus of experts outside Japan, where the disaster context is very different. Additional surveys with more detailed questionnaires with clearer aims and goals in this category, and comparisons with the evaluation of acute-phase activities as a control, will be required.

## 5. Conclusions

This study presented the preliminary scope for consensus on MHPSS approaches after natural hazard events among mental health experts in Japan. Participants were, for the most part, in agreement regarding focuses and required preparedness and actions for long-term support. There was still low consensus especially on defining the timeframe “long-term.” The acute phase and long-term phase were identified as being different in dimension rather than category in this survey. This is the first attempt to investigate experts’ consensus view on long-term MHPSS after a natural hazard event. Despite the limitations of the representativeness due to the biases associated with online surveys in Japanese, our findings will provide important scientific evidence for the development of future plans for a qualitative improvement in long-term MHPSS. Other countries and regions with different disaster situations will probably have different consensus ranges. Therefore, conducting similar studies in other countries/regions may be beneficial for developing national approaches and guidelines for natural hazard event response and preparedness.

## Figures and Tables

**Table 1 ijerph-19-03022-t001:** Demographic data of participants.

Characteristics	*n* (%)
(a) Workplace of participants	
National/local municipality or their agency for health service provision	68 (22.3%)
Medical institute (e.g., hospital, clinic)	153 (50.2%)
Educational and/or research institute (e.g., university)	35 (11.5%)
Elsewhere	46 (15.1%)
N/A (e.g., retired)	3 (1.0%)
Total	305 (100%)
(b) Occupation of participants	
Medical doctor	50 (16.4%)
Community (public health) nurse	30 (9.8%)
Nurse	50 (16.4%)
Psychosocial worker	37 (12.1%)
Social worker	5 (1.6%)
Psychologist	92 (30.2%)
Teacher	7 (2.3%)
Other occupation	34 (11.1%)
Total	305 (100%)
(c) Location of participants	
Hokkaido, Tohoku	46 (15.1%)
Kanto	116 (38.0%)
Chubu, Hokuriku	37 (12.1%)
Kansai	33 (10.8%)
Chu-shikoku	25 (8.2%)
Kyushu, Okinawa	43 (14.1%)
N/A (e.g., out of Japan)	5 (1.6%)
Total	305 (100%)

**Table 2 ijerph-19-03022-t002:** Descriptive statistics of the survey.

No.	Items	Mean	SD	Min	Max	n
Category A: Terminology setting-definition of “long-term”						
1	Acute stress disorder or traumatic response ends as a natural recovery process (about 1–2 months)	2.45	1.79	0	5	303
2	Natural disaster event is over and no further serious damage is anticipated anymore	2.72	1.79	0	5	302
3	Fundamental infrastructures for basic livelihood are recovered	3.05	1.58	0	5	301
4	The local mental health facilities are recovered and do not need to rely on external support anymore	3.53	1.35	0	5	302
5	Support based on Disaster Relief Act is over	2.55	1.63	0	5	298
6	Transition from staying at an evacuation site to living in temporary houses starts	2.90	1.70	0	5	299
Category B: Priority in activity for long-term mental health support						
7	Diagnostic evaluation and treatment for psychological disorders	2.97	1.43	0	5	302
8	Education for families and communities to reduce stress and promote recovery	4.10	0.99	1	5	301
9	Case work and outreach for families and communities	4.12	1.01	0	5	301
10	Collaboration between mental healthcare providers and national/local municipalities	4.34	0.85	1	5	302
11	Collaboration between mental health-care providers and physical healthcare providers	3.97	1.01	1	5	301
12	Broadly opening the door for consultation by disaster survivors including non-disaster-related issues	3.54	1.36	0	5	299
13	Capacity building for mental health management using a standardized training program such as Psychological First Aid (PFA)	3.83	1.19	0	5	302
14	Support for evacuees who moved to another community or region due to the disaster event	3.82	1.13	0	5	300
15	Maintain special medical care system for severe stress disorder such as PTSD and grief	3.93	1.08	0	5	303
Category C: System and preparedness for better support						
16	Establishing a local mental healthcare center dedicated to long-term support, when the damage of the disaster is severe.	4.12	1.14	0	5	301
17	Setting standardized criteria for the establishment of a mental health-care center	3.85	1.19	0	5	303
18	Expansion of existing capacity of healthcare providers and local municipalities, rather than establishing a new facility	3.74	1.24	0	5	301
19	Development of a norm and standard for organizational provision of training and guidance for long-term mental health support	4.44	0.81	1	5	303
20	Large scale provision of training programs for capacity building on long-term mental health support	4.46	0.80	1	5	303
Category D: Transition from acute support to long-term support						
21	Acute support focuses on medicine, while long-term support focuses on broader mental health activities including social support. Therefore, these two phases of support should be organized separately and long-term support should be initiated clearly after the termination of acute support.	2.19	1.51	0	5	300
22	Mental and social support should be provided from immediately after a disaster event as it is important for the long-term outcome of disaster survivors. Therefore, acute and long-term support should be organized in parallel and in collaboration with each other rather than dividing them by chronological order.	4.28	1.02	1	5	300
Category E: Actions to improve preparedness for future disasters						
23	Technical support and advice for future disaster areas by local municipalities, local mental health center and health-care workers who have experienced past major disasters	3.86	1.13	0	5	301
24	Training of long-term mental health support for healthcare workers engaged in acute response	4.51	0.74	2	5	303
25	Accumulation and review of different kinds of literatures regarding long-term support for evidence-based capacity building of specialists and development of expert network	4.28	0.83	2	5	303
26	Broadly accumulate and inherit past experience for disaster responses.	4.51	0.74	2	5	304
27	Assessment of the impact of activities in past disasters through specific survey	4.18	0.97	0	5	303
28	Enhance support and mental health security for workers in a disaster area	4.46	0.76	1	5	303

SD: standard deviation.

## Data Availability

Not applicable.

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
