# Peer review of "Long-Term Mental Health Support after Natural Hazard Events: A Report from an Online Survey among Experts in Japan"

_ijerph, 2022, doi:10.3390/ijerph19053022_

Round 1
Reviewer 1 Report
The article is extremely interesting and deals with an important issue. Its structure and the aim are appropriate. Only the conclusions need to be expanded - now is it only a summary. I propose to add a limitation of research, and to indicate the research directions for the future in more detail.
Reviewer 2 Report
First of all according to UNDER decision that there is no such thing as a natural disaster. Please avoid it and you other term (there are natural hazards, and there are disasters).
There is no section on ethical considerations of the study.
It is suggested to compare the results of the present research with some similar studies which is done before.
The study has several both limitations and strengths, that (I believe) are simply not stated, except for some parts of the document. Authors must put an additional effort on identifying, describing and discussing these issues.
Author Response
Please see the attachment. Thank you.

This manuscript is a resubmission of an earlier submission. The following is a list of the peer review reports and author responses from that submission.
Round 1
Reviewer 1 Report
The literature review needs to be expanded, without focusing only on Japan
The demographic data are very limited in terms of the information contained therein
Although I would replace table 1 with table 2 - the certificate should be with the methods and the table with questions and their summary in the results
The results are far too little described
The conclusions should also be expanded, among others in terms of the usefulness of the applied research and the possibility of their further development in the future
Reviewer 2 Report
This paper aims to provide preliminary evidence of the degree of consensus on the approach to long-term mental health activities after a disaster. However, results lost to reach consensus about 'long term'. It is obvious to lead such a conclusion without consideration for difference of damage in each disaster affected area. The paper is a simple survey with little scientific findings.